# Continuous detection of Chikungunya Virus in a passive surveillance system in southern Thailand, 2012–2019

**Aaron Farmer**[1]* , **Kathryn B. Anderson**[1,2] , **Darunee Buddhari**[1] ,
**Thanaporn Hortiwakul**[3] , **Boonsri Charernmak**[3] , **Butsaya Thaisomboonsuk**[1] ,
**Tippa Wongstitwilairoong**[1] , **Taweewun Hunsawong**[1] , **Chonticha Klungthong**[1] ,
**Piyawan Chinnawirotpisan**[1] , **Sarunyou Chusri**[3] , **Stefan Fernandez**[1]

1 Department of Virology, Armed Forces Research Institute of Medical Sciences, Bangkok, Thailand,
2 SUNY Upstate Medical University, Syracuse, New York, United States of America, 3 Prince Songkla
University, Songkhla, Thailand

☯ These authors contributed equally to this work.
* dr.arfarmer@gmail.com

## Abstract

Chikungunya virus (CHIKV) infections are distributed across the globe, causing significant
and, often, lasting morbidity. CHIKV vaccines are in development, but their evaluation is limited by the unpredictability of CHIKV transmission, which classically manifests as explosive
epidemics separated by variable interepidemic periods. A passive surveillance study for
undifferentiated febrile illness was established in southern Thailand in 2012 and is ongoing.
Among 1466 febrile individuals with acute and convalescent specimens, 398 (27.1%) had
molecular or serological evidence of acute CHIKV infection. The proportions of participants
confirmed to have CHIKV infection differed by year, being highest during epidemic periods
(41.1% in 2018–2019, corresponding to a large regional CHIKV outbreak, compared to
19.3% in 2012–2017). These data suggest persistent circulation of CHIKV in the study area,
though additional studies are needed to confirm these findings and to discern whether this
persistence reflects widespread, low-level transmission or migrating bursts of focal epidemic
activity.

## Author summary

Chikungunya virus (CHIKV) infections found worldwide and can lead to significant
long-lasting morbidity due to chronic joint pain. Although CHIKV vaccines are in development (with one recently receiving accelerated approval based on a surrogate endpoint), the transmission patterns of CHIKV are unpredictable, presenting as explosive
epidemics separated by variable interepidemic periods, making clinical trials difficult. In
southern Thailand, we monitored acute febrile illnesses via an enhanced passive surveillance system from 2012 to 2019 and by combining molecular and serological methods,
acute CHIKV was identified in 27.1% of 1466 individuals. These findings suggest

pntd.0012776

Faculty of Medicine, MALAYSIA

**Data Availability Statement:** The data that support
the findings of this study have been made publicly
available from figshare with the identifier(s):

10.6084/m9.figshare.27079927 Direct link: https://figshare.com/articles/dataset/Southern_Thailand_Passive_Surveillance_Dataset_for_Sharing_21_Sep_2024_data_csv/27079927 The sequences generated from this study have been submitted to GenBank under the accession numbers MN974206-208, MN974211, MN974214-224, and ON262791-796.

**Funding:** This study was funded by the Armed Forces Health Surveillance Branch (AFHSB) and its Global Emerging Infections Surveillance (GEIS) Section, USA under grant numbers: P0086_21_AF; P0128_20_AF; P0108_19_AF; P0022_22AF. The funders had no role in study design, data collection and analysis, decision to publish, or preparation of the manuscript.

**Competing interests:** The authors have declared that no competing interests exist.

persistent circulation of chikungunya in the region and that there may be unknown reservoirs or low-level transmission occurring outside of detected outbreaks.

## Introduction

Chikungunya virus (CHIKV) is an emerging alphavirus, distributed across tropical and subtropical regions of Africa, Asia, and the Americas [1]. It is spread by *Aedes albopictus* and *Aedes aegypti* mosquitoes, thus there is considerable geographic overlap between endemic regions for CHIKV and other endemic or emerging arboviruses, notably dengue virus (DENV) and Zika virus (ZIKV). CHIKV was first described in the 1950s in Tanzania and sporadic epidemics occurred in Asia and Africa in the following decades [2]. This included large outbreaks of CHIKV in Thailand in the late 1950s and early 1960s, with an estimated 31% of Bangkok becoming infected in a large epidemic in 1962 [3]. The epidemiology of CHIKV changed beginning in 2004, with simultaneous geographic expansions of two different CHIKV genotypes, the East Central South African genotype (ECSA) and Asian genotypes, causing novel, repeated, large epidemics in urban centers across the globe. The Indian Ocean Lineage (IOL) of the ECSA genotype was first reported in Kenya in 2004 [4] and was subsequently associated with large epidemics in south and southeast Asia and more temperate climates, attributed to an envelope glycoprotein mutation, E1-A226V, conferring increased transmissibility in *Aedes albopictus* mosquitoes [5,6]. The same genotype with the E1-A226V mutation was responsible for outbreaks in Thailand in 2008–2009 and 2013 [7]. In 2018–2019, a large outbreak of CHIKV infection in Thailand, defined as a new suspected case in at least 2 patients or more within 2 weeks and at least one confirmed CHIKV, was reported with ECSA as the primary genotype [8]. Notably, the dominant strain for this outbreak strain did not contain the E1-A226V mutation, but rather two mutations including E1-K211E and E2-V264A which are associated with virus adaptation in *Aedes aegypti* leading to improved vector competency [9].

CHIKV is characterized by an abrupt onset of high fever, profound joint pain, and rash [10]. Symptoms typically resolve within 1–2 weeks, but it is estimated that in 30–40% of infected individuals, CHIKV-related joint pain may recur and may persist for years [11]. Recently, the first vaccine for chikungunya (IXCHIQ) has been approved but is not currently available in southeast Asia and treatment is otherwise supportive. Critical illness and deaths have been reported with CHIKV, typically associated with exacerbations of underlying comorbidities in older individuals but also related to neurological manifestations of CHIKV such as Guillain-Barre syndrome and CHIKV encephalitis [12]. It is often reported that the majority of CHIKV infections are symptomatic, perhaps as high as 70–90% [13,14]. However, a study performed in the Philippines, an area with multiple prior CHIKV epidemics, found a low symptomatic infection rate of 18% [15]. Thus, the true symptomatic to asymptomatic ratio for CHIKV, and risk factors for acute and chronic disease, are variable based on differences in populations, risk factors and methodologies [16].

There are multiple gaps in our understanding of CHIKV epidemiology, which pose challenges for the development and evaluation of CHIKV vaccines. CHIKV epidemics are frequently explosive, emerging unpredictably and with transmission typically waning to low levels after high levels of population immunity are achieved. For example, a large CHIKV outbreak in Reunion Island in 2005 was estimated to have infected 32% of the population [17,18]. Seroprevalence to CHIKV in Guadeloupe, the Caribbean, after an initial epidemic in 2015, was estimated to be 48.1% [19]. Recurrent epidemics of CHIKV do occur once the virus has been

established in a region (e.g., Nicaragua [20] and the Philippines [21]) but these repeat epidemics occur unpredictably, typically after a period of several years. It remains unclear whether, during these inter-epidemic periods in affected countries, CHIKV retreats to its sylvatic origins, cycling among non-human primate populations and sylvatic mosquitoes [22], or whether it continues to circulate at low levels of undetected endemicity in human populations. Thus, despite the recognized need for efficacious CHIKV vaccines [23], plans for efficacy trials have been hampered by the apparent unavailability of sites with long-standing and reliable CHIKV transmission [24]. Despite this potential obstacle, multiple CHIKV vaccines are in development [25], with one recently receiving US FDA approval as mentioned above [26].

In this manuscript, we present data from an ongoing passive fever surveillance study in southern Thailand that indicates possible persistent circulation of chikungunya virus in the region. These findings suggest that there are areas of the world where transmission of CHIKV may be more consistent, yielding important sites for research into CHIKV transmission and pathogenesis, and providing opportunities for countermeasure development and evaluation.

## Methods

### Ethics statement

This study was approved by Office of Human Research Ethics Committee, Prince of Songkhla University, Institutional Review Board for the Protection of Human Subjects-State University of New York (SUNY) Upstate Medical University, and Walter Reed Army Institute of Research Human Subject Protection Branch (protocol #1934).

### Study site

Songkhla province, in southern Thailand, is located at a critical zone of deforestation and urbanization near the border with Malaysia (Fig 1). This area is of particular interest for surveillance of emerging arboviral diseases. For example, in 2013, this site provided valuable early evidence of Zika virus circulation in Thailand [27]. Large chikungunya outbreaks were detected and confirmed in southern Thailand in 2008–9 [7] and 2018–19 [28]. CHIKV incidence data from the Thai Ministry of Public Health (MoPH) support a persistent burden of CHIKV infection in southern Thailand, with multiple southern provinces (including Songkhla) reporting CHIKV cases during the 'interepidemic' years of 2011–2017, though incidence rates were low during this period (S1 Fig).

### Surveillance platform

In 2012, the Armed Forces Research Institute of Medical Sciences (AFRIMS), Department of Virology, began an ongoing collaboration with Prince of Songkhla University (PSU) in Songkhla, Thailand, to identify causes of undifferentiated febrile illness in the region. PSU is the largest medical school in southern Thailand and operates a major public hospital, Songklanagarind Hospital, located in the largest city in Songkhla, Hat Yai. This hospital is a 1000 bed tertiary care hospital, which receives referrals from multiple onsite and offsite outpatient clinics.

Individuals of all ages presenting acutely to participating clinical centers with undifferentiated fever or history of fever within 7 days, not attributable to any non-arboviral cause of fever such as pneumonia or urinary tract infection, were eligible for enrollment. Demographic data including sex, age, occupation, direct and indirect contact with animals and housing conditions were included. Written informed consent/assent was obtained per the primary protocol, with parental/guardian consent and participant assent for children per local requirements.

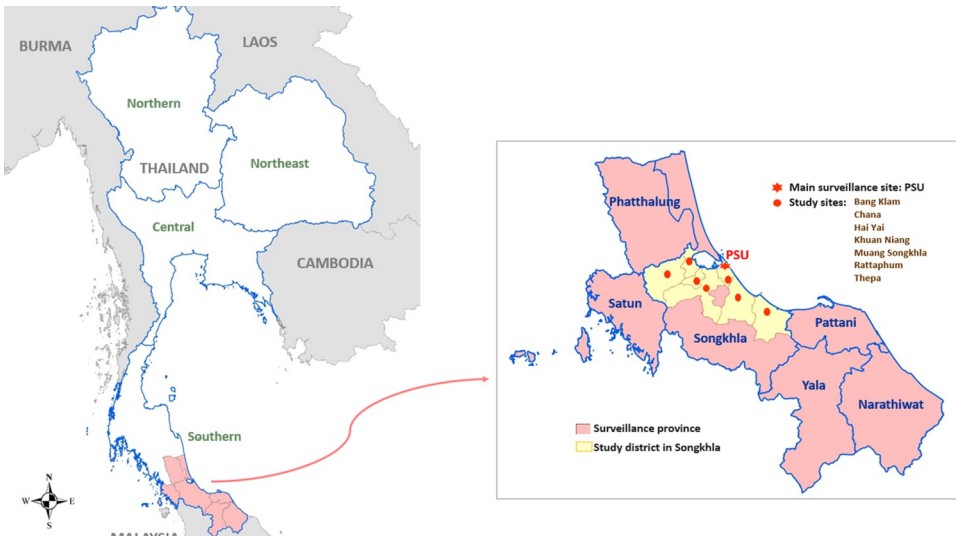

**Fig 1. Map of Thailand, highlighting study locations within the southern region (in pink).** Inset shows locations of the main surveillance site (PSU; red star) and 7 associated study sites (red dots) within Songkhla province (in yellow). Basemap shapefile downloaded from GADM at https://gadm.org and GADM license allowing use is located at https://gadm.org/license.html.

Patients meeting criteria for testing had 5 mL of blood collected at their acute (enrollment) visit and were requested to return no more than 21 days later for a repeat (convalescent) sample. Those not meeting these criteria or only having a single acute specimen were not included in the analysis. Acute specimens underwent RT-PCR testing for DENV, CHIKV, and ZIKV, followed by serological testing for these same arboviral infections as the volume of residual specimen allowed. Patients included in this analysis were enrolled between June 2012 and December 2019 at Songklanagarind Hospital and 7 community clinics that utilize PSU Hospital as a referral center (Figs 1 and 2).

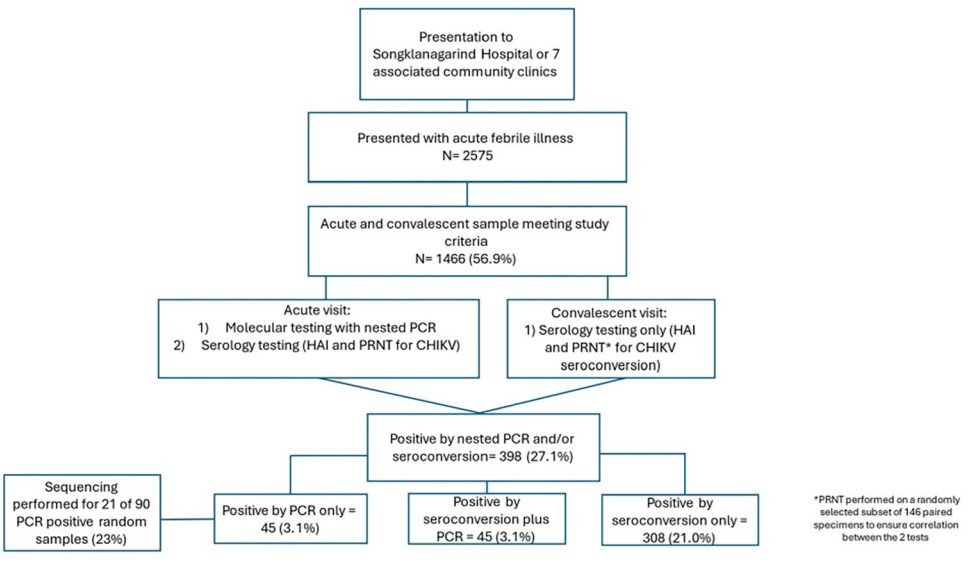

**Fig 2. Overview of study enrollment.**

## Confirmation of CHIKV infection

Molecular and serological methods were applied to confirm acute CHIKV infections in the cohort. Qualitative detection of CHIKV RNA was performed using a nested RT-PCR method adapted from Porter et al [29], which detects the CHIKV capsid gene. A hemagglutination inhibition (HAI) assay, using antigens obtained from suckling mouse brain infected with a CHIKV Ross strain isolate, was applied to quantify CHIKV antibody titers in acute and convalescent serum specimens [30]. The inverse of the highest dilution of serum that inhibited hemagglutination was defined as the HAI titer. Comparing paired acute and convalescent samples, seroconversion to CHIKV was defined as an increase in HAI titers from undetectable ($<10$) to $>= 40$ or, for those with CHIKV titers $>= 10$ in acute specimens, a four-fold rise in titers. HAI was chosen over ELISA to allow assessment of functional antibody response to CHIKV.

## Neutralizing antibody testing

Neutralizing antibody titers to CHIKV were measured in a randomly-selected subset of acute febrile illness specimens from study participants using a standard plaque reduction neutralization test (PRNT) in LLC-MK2 cells to reference CHIKV vaccine strain 181 clone 25. Serial dilutions of serum were incubated with a fixed infectious dose of virus and the 50% plaque reduction neutralization titer ($PRNT_{50}$) calculated using log probit analysis [31].

## CHIKV genomic sequencing

CHIKV RT-PCR positive samples were amplified by 3 passages in C6/36 cells. RNA was extracted from the culture supernatant using QIAamp Viral RNA Mini Kit (QIAGEN). Whole genome amplification was performed using SuperScript III One-Step RT-PCR with Platinum Taq Hi-Fidelity (Invitrogen) and CHIKV specific primers. DNA libraries were constructed and sequenced using QIASeq FX DNA library preparation (QIAGEN) and MiSeq (2x250 cycles, Illumina) kits, respectively [32]. Total reads were mapped to CHIKV reference genome (MK848202 Thailand 2018) using v 1.4.2, a WRAIR Viral Disease Branch (VDB) bioinformatics pipeline ngs_mapper.

## Statistical methods

Statistical analyses were restricted to individuals with paired acute and convalescent sera, with the convalescent specimen collected at least 7 days after the acute specimen, and to those who had sufficient volumes of specimen remaining for CHIKV serological testing. A confirmed acute CHIKV infection was defined as detection of CHIKV RNA by RT-PCR in an acute specimen or seroconversion to CHIKV by HAI. Study periods were divided into interepidemic and epidemic periods, based upon the relevant known periods of epidemic CHIKV activity in southern Thailand (i.e., 2018–2019). Correlations between CHIKV $PRNT_{50}$ and HAI log antibody titers in convalescent specimens were assessed using Spearman's rank correlation rho statistic. Statistical analyses were conducted in IBM SPSS (v26, 2019) and R (R Core Team, 2014) and figures were produced using the package ggplot2 [33].

# Results

## Overview of study participants

From 2012 to 2019, a total of 2575 individuals presented with febrile illness to participating clinical sites in Songkhla province, Thailand, and provided acute illness specimens. Of these, 1466 (56.9%) had both an acute and a convalescent specimen available, collected within an appropriate follow-up interval, and with sufficient volumes of specimen for CHIKV serological

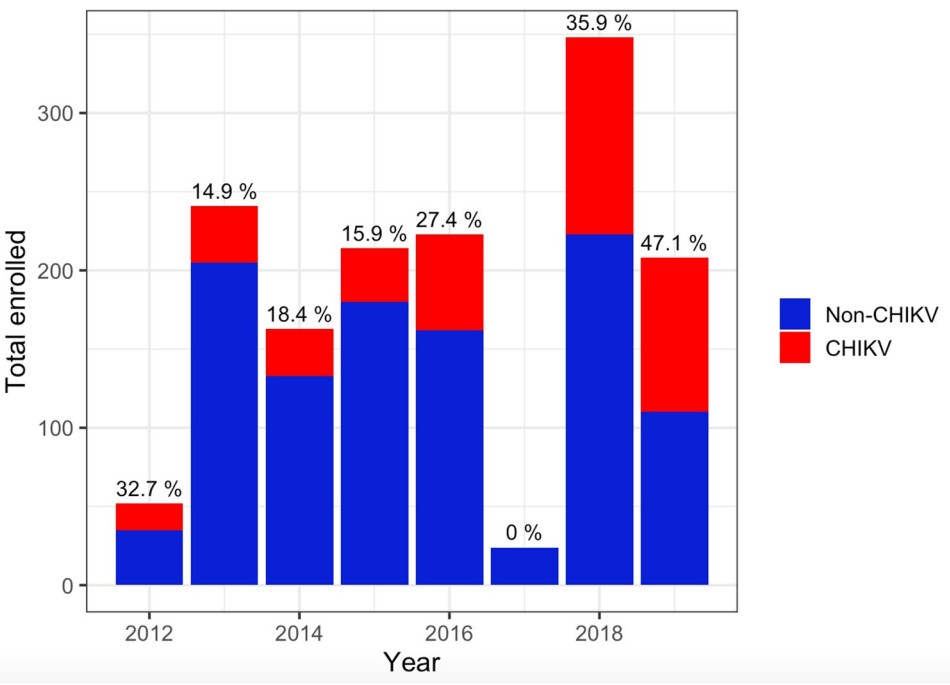

**Fig 3. Total participants enrolled by year, with red shading indicating numbers of participants confirmed to have acute chikungunya infection (% indicates proportion of total infected).** Confirmation of CHIKV infection was by detection of viral RNA through nested RT-PCR or identification of a at least four-fold rise in CHIKV HAI antibody titers between paired specimens.

testing. Enrollment varied by year, with peak enrollment in 2013, 2015, and 2018 (Fig 3). The median age of enrolled participants was 39 years (range: 14–87) and 48.6% were female (Table 1). The majority were agricultural workers (76.3%) who were largely employed in rubber plantations. Participants resided in multiple provinces within southern Thailand, consistent with the broad catchment area of the clinical sites. The majority of participants resided within Songkhla province, reflecting both the higher population density of this province, containing the two urban centers of Hat Yai and Muang Songkhla, and proximity to the primary clinical sites. The median amount of time from illness onset to acute specimen collection was 3 days (range: 1–7). The median time interval between acute and convalescent specimen collection was 13 days (range: 7–21). All subjects reported fever (100%), as this was part of the inclusion criteria for enrollment. Headaches (44.3%), and myalgia (42.8%) were the next two most predominant symptoms.

## Chikungunya incidence and features

Among the 1466 individuals that presented with febrile illness from 2012–2019 and were eligible for inclusion in this analysis, 398 (27.1%) had confirmed acute CHIKV infection by molecular and/or serological methods (PCR and/or HAI positive by criteria described); of these 45 were positive for both PCR and seroconversion (3.1%), 45 were PCR positive alone (3.1%) and 308 were positive by seroconversion alone (21.0%) (Fig 2).

Individuals that seroconverted but were RT-PCR negative presented later in the course of their illness, at 2.64 days post-onset of symptoms compared to 2.49 days for RT-PCR+ individuals (p<0.01 by ANOVA, Table 2). Convalescent HAI titers to CHIKV were higher for those that were RT-PCR negative (S2A Fig), as was the magnitude of the rise in HAI antibody titers

**Table 1. Characteristics of enrolled participants overall and with and without evidence of acute CHIKV infection.**

| | | Overall | CHIKV-confirmed | Non-CHIKV | p-value* |
|---|---|---|---|---|---|
| Total | | 1466 | 398 (27.1%) | 1068 (73.0%) | |
| Gender | | | | | |
| | Male | 753 (51.4%) | 210 (27.9%) | 543 (72.1%) | 0.513 |
| | Female | 713 (48.6%) | 188 (26.4%) | 525 (73.6%) | |
| Age | | | | | |
| | 10–19 | 109 (7.4%) | 23 (21.1%) | 86 (78.9%) | 0.082 |
| | 20–39 | 705 (48.1%) | 200 (28.4%) | 505 (71.6%) | |
| | 40–59 | 468 (31.9%) | 136 (29.1%) | 332 (70.9%) | |
| | 60+ | 184 (12.7%) | 39 (21.2%) | 145 (78.8%) | |
| Occupation (limited to those aged > = 18 years) | | | | | |
| | Agricultural worker | 1114 (76.1%) | 308 (27.6%) | 806 (72.4%) | 0.478 |
| | Non-agricultural | 350 (23.9%) | 90 (25.7%) | 260 (74.3%) | |
| Province | | | | | |
| | Songkhla | 1026 (70.0%) | 253 (24.7%) | 773 (75.3%) | 0.031 |
| | Pattani | 169 (11.5%) | 56 (33.1%) | 113 (66.9%) | |
| | Yala | 88 (6.0%) | 30 (34.1%) | 58 (65.9%) | |
| | Narathiwat | 78 (5.3%) | 22 (28.2%) | 56 (71.8%) | |
| | Satun | 70 (4.8%) | 26 (37.1%) | 44 (62.9%) | |
| | Phatthalung | 35 (2.3%) | 11 (31.4%) | 24 (68.6%) | |
| Symptoms (% reporting each symptom)** | | | | | |
| | Fever | 1466 (100%) | 398 (100%) | 1068 (100%) | — |
| | Headache | 650 (44.3%) | 176 (44.2%) | 474 (44.4%) | 0.959 |
| | Arthralgia | 381 (26.0%) | 112 (28.1%) | 269 (25.2%) | 0.252 |
| | Myalgia | 627 (42.8%) | 202 (50.8%) | 425 (39.8%) | <0.01 |
| | Rash | 278 (19.0%) | 75 (18.8%) | 203 (19.0%) | 0.943 |
| | Chills | 293 (20.0%) | 89 (22.4%) | 204 (19.1%) | 0.165 |
| | Abdominal Pain | 68 (10.3%) | 15 (13.2%) | 53 (9.7%) | 0.262 |

* p- value obtained through Chi-squared testing

** 100% of participants reported fever since the presence of fever was one of the inclusion criteria for the study. Abdominal pain data were available only for a subset of 663 participants.

between acute and convalescent (S2B Fig), supporting differences in the timing of sampling relative to onset of infection for those confirmed by molecular versus serological methods. The incidence of confirmed infection varied from year to year, with highest incidence in 2012 (32.7%), 2018 (37.6%), and 2019 (47.0%) (Fig 2). Febrile adults aged 20–39 years and 40–59 years were noted to present with CHIKV at a higher percentage compared to febrile children/teenagers and older adults, although not at a statistically significant rate (28.4% and 29.1% versus 21.1% and 21.2%, respectively, p = 0.08) (Table 1). Similar proportions of males and females were confirmed to have acute CHIKV infection. The rates of seroconversion in agricultural and non-agricultural workers were 27.6% and 25.7% (p = 0.47). The proportion of individuals confirmed to have acute CHIKV differed significantly by province, with the highest proportions observed in Pattani, Yala, and Satun (p = 0.03). Reported symptoms did not differ for febrile individuals with and without confirmed CHIKV, except for myalgia, which was significantly different between the two groups (p<0.01). Notably, only 28.1% of CHIKV-confirmed individuals reported arthralgia.

**Table 2. Characteristics of participants during epidemic and inter-epidemic periods.**

| | | Overall | Epidemic (2018–2019) | Inter-epidemic (2012–2017) | p-value |
|---|---|---|---|---|---|
| **Total participants** | | 1466 | 528 (36.0%) | 938 (64.0%) | - |
| CHIKV-confirmed | | 398 (27.1%) | 217 (41.1%) | 181 (19.3%) | <0.01 |
| **CHIKV-confirmed participants** | | | | | |
| Gender | | | | | |
| | % Female | 188 (47.2%) | 96 (44.2%) | 92 (50.8%) | 0.190 |
| Age | | | | | |
| | Mean age (SD) | 37.99 (15.14) | 37.4 (15.84) | 38.70 (14.26) | 0.014 |
| Occupation | | | | | |
| | % Agricultural worker | 308 (77.4%) | 175 (80.6%) | 133 (73.5%) | 0.089 |
| Clinical | | | | | |
| | Fever | 398 (100%) | 217 (100%) | 181 (100%) | - |
| | Headache | 176 (44.2%) | 95 (43.8%) | 81 (44.8%) | 0.846 |
| | Arthralgia | 112 (28.1%) | 70 (32.3%) | 42 (23.2%) | 0.045 |
| | Myalgia | 202 (50.8%) | 139 (64.1%) | 63 (34.8%) | <0.01 |
| | Rash | 75 (18.8%) | 39 (18.0%) | 36 (19.9%) | 0.626 |
| | Chills | 89 (22.4%) | 55 (25.3%) | 34 (18.8%) | 0.118 |
| Days from symptom onset to presentation | | | | | |
| | Mean days (SD) | 2.59 (1.15) | 2.44 (1.00) | 2.77 (1.29) | <0.01 |
| RT-PCR | | | | | |
| | %PCR positive | 90 (22.6%) | 86 (39.6%) | 4 (2.2%) | <0.01 |
| HAI seroconversion | | | | | |
| | %HAI seroconversion | 353 (88.7%) | 175 (80.6%) | 178 (98.3%) | <0.01 |
| **CHIKV by HAI seroconversion** | | | | | |
| Days from symptom onset to presentation | | | | | |
| RT-PCR negative: Mean days (SD) | | 2.64 (1.21) | 2.46 (1.05) | 2.77 (1.31) <0.01 | |
| RT-PCR positive: Mean days (SD) | | 2.49 (0.97) | 2.50 (0.98) | - | |

\* p- value obtained through Chi-squared testing

The majority of participants were enrolled during the inter-epidemic period of 2012–17 (938 of 1466 individuals, or 64.0%, Table 2). The incidence of confirmed CHIKV infection was higher during the epidemic period of 2018–19 (41.1% versus 19.3% in the inter-epidemic period, p<0.01). A higher proportion of infections were RT-PCR-confirmed during the epidemic period (16.3 versus 0.4%), during which time febrile participants presented sooner after onset of symptoms (2.46 days from symptom onset to presentation and enrollment versus 2.77 days p<0.01).

There was no difference in the age, gender, or occupation of individuals confirmed to have acute CHIKV infection in epidemic versus inter-epidemic periods. The distribution of cases between epidemic and inter-epidemic periods was different by province; for the largest provinces of Songkhla and Narathiwat, the majority of cases were identified in the epidemic period (S1 Fig). The remaining four provinces had higher numbers of cases confirmed within the cohort during the epidemic period. For every age group, individuals presenting with febrile illness were more likely to be CHIKV-infected during the epidemic period, with age-specific patterns that roughly paralleled the inter-epidemic period (Fig 4). A total of 398 CHIKV cases had clinical data for evaluation. Of these, only myalgia was significantly different with 139 (64.1%)

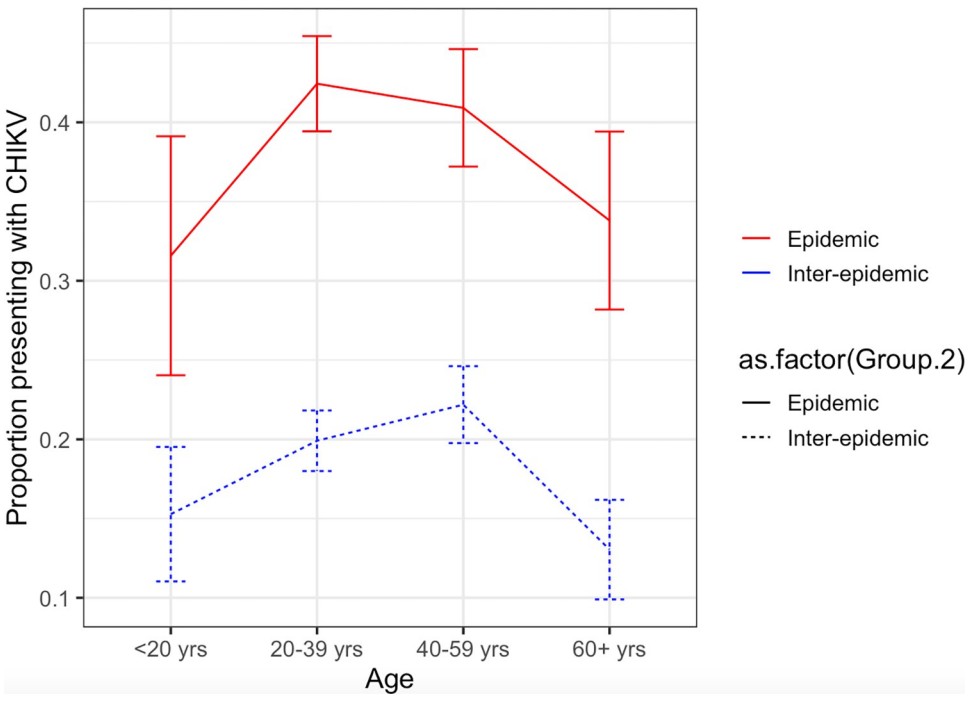

**Fig 4. Proportion of individuals presenting with febrile illness confirmed to have acute CHIKV infection by age, contrasting epidemic periods (2018–2019; red) and inter-epidemic periods (2012–2017; blue).** 95% confidence intervals are shown.

vs 63 (34.8%) reported between the epidemic and inter-epidemic period. No differences were noted in rates of headache (p = 0.846), arthralgia (p = 0.045), rash (p = 0.626) or chills (p = 0.118) (Table 2).

## Correlations between CHIKV HAI and CHIKV neutralizing antibody results

HAI was used as the primary screening assay on all samples with CHIKV neutralizing antibody testing on a randomly selected subset of paired specimens from 146 individuals. Comparing convalescent antibody titers to CHIKV identified by PRNT50 and HAI, there was significant correlation between convalescent titers obtained between the two assays (Spearman's rank correlation rho of 0.817, p<0.01; S3 Fig). The specificity of HAI to detect a seroconversion, compared to PRNT50, was 95.5% and sensitivity to detect a seroconversion was 87.0%.

## Annual Patterns of CHIKV and DENV

Given the common vector and seasonal overlap, we also analyzed acute dengue infection as a marker of general arboviral transmission in the area. Acute DENV infection was confirmed by RT-PCR in 259 (17.7%) of all participants presenting with acute febrile illness. Among these, 3 were also RT-PCR-positive for CHIKV, for a PCR-confirmed coinfection rate of 1.2%. All four serotypes were identified as circulating in the study area, with the dominant serotypes being DENV-3 (87 confirmed infections, or 33.6% of all RT-PCR-confirmed DENV infections) and DENV-4 (74 confirmed infections, 28.6%). Seasonal patterns of DENV and CHIKV detection generally overlapped, though in 2018 an early peak was observed for DENV and CHIKV,

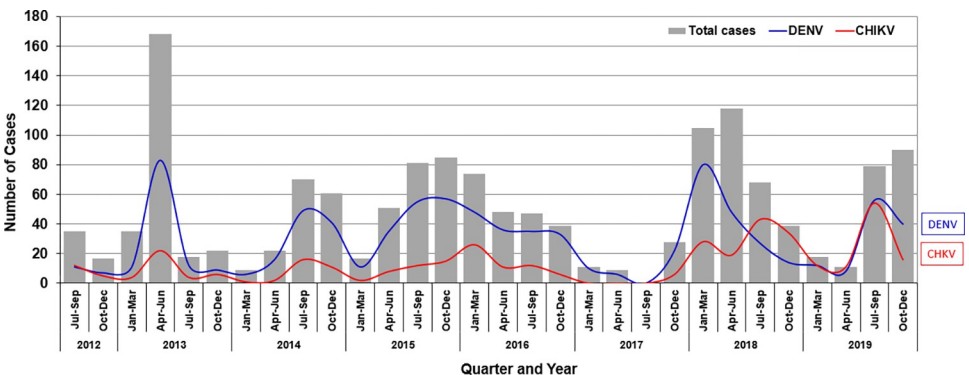

**Fig 5. Annual incidence and seasonality of confirmed DENV (blue) and CHIKV (red) infections among enrolled participants presenting with acute febrile illnesses to the study sites in Southern of Thailand, 2012–2019.**

followed by a larger peak for CHIKV later that summer (which corresponded to a large national CHIKV epidemic) (Fig 5).

## Genetic characterization

Twenty-one CHIKV viruses were successfully isolated and sequenced from participants in the study from 2012–2019; all belonged to the IOL of the ECSA strain (Fig 6). Among these, two CHIKV isolates from 2012 retained E1-A226V mutations and clustered with other Thai sequences from 2009–2013. Thai CHIKV during this inter-epidemic period (2007–2013) were interspersed with those from other South and Southeast Asian CHIKV. The other 19 CHIKV sequences from 2018 (11 sequences), and 2019 (8 sequences) were clustered with other Thai

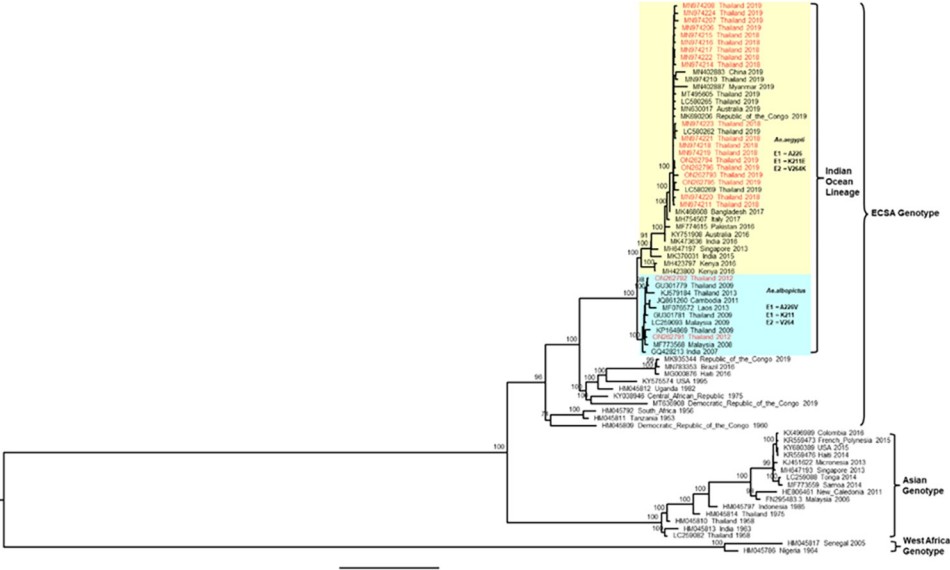

**Fig 6. Maximum-likelihood tree of CHIKV Coding Sequences.** The maximum-likelihood tree of CHIKV coding sequences (CDS) 11,216 bp included 21 sequences from this study labelled in red, and 54 sequences from GenBank labelled in black. The tree was constructed using IQ-Tree with substitution model GTR+F+G4 and 1,000 ultrafast bootstrap replicates. Amino acid mutations associated with vector adaptation were indicated. The sequences generated from this study have been submitted to GenBank under the accession numbers MN974206-208, MN974211, MN974214-224, and ON262791-796.

CHIKV reported in the large epidemic of 2018–2019; representing expansion of a clade previously found circulating in Africa, South Asia, and Singapore beginning in 2015. Thai CHIKV isolates from this period possessed E1-K211E, E1-I317V, E2-G205S, and E2-V264A mutations but no mutations at E1-226 position (E1-226A).

## Discussion

This study provides evidence of possible long-term persistence of CHIKV transmission in an area hyperendemic for arboviral diseases in southern Thailand. High levels of CHIKV transmission were identified, with 27.1% of patients presenting with undifferentiated fever from 2012–2019 confirmed to have acute CHIKV infection. Most noteworthy, the incidence of acute CHIKV during inter-epidemic periods of the study (2012–2017) was higher than expected, at 19.3%. These high frequencies may reflect the heightened sensitivity of combined molecular and serological methods with paired sera and / or a uniquely permissive environment for CHIKV transmission in the study area. If confirmed, these findings offer promise for the development and evaluation of much-needed countermeasures for CHIKV such as vaccines, antivirals, and diagnostics, which have historically struggled to find appropriate sites for trials given the sporadic and unpredictable nature of CHIKV epidemics.

This is not the first report of long-term persistence of CHIKV between epidemic periods. In northeastern Thailand during the inter-epidemic period of 2016–17, 4.9% of acute febrile patients had detectable CHIKV RNA by RT-PCR and 10.6% had detectable anti-CHIKV IgM and IgG [34]. A cohort study of factory workers in inter-epidemic periods in Bandung, Indonesia, found a CHIKV incidence rate of 10.1/1,000 person years based upon serological testing of longitudinal specimens [35]. Using molecular methods (RT-PCR) for CHIKV confirmation only, a pediatric febrile illness surveillance study in Kenya found that CHIKV accounted for 12.7% of acute illnesses, from 2014–2018 [36]. Notably, among 443 children who were RT-PCR-positive for CHIKV infection, 170 had a subsequent acute febrile illness with 19 of these also confirmed to be CHIKV by RT-PCR during the study period. These subsequent CHIKV infections occurred after a range of 2–43 months after the first detected infection. For all apparently CHIKV-endemic regions, including our study site in southern Thailand, it remains unclear whether this long-term persistence reflects low-level, widespread transmission or focal, migratory epidemics. Supporting the latter scenario, a highly focal epidemic was described in Brazil in 2017, occurring 2.5 years after the nationwide epidemic and restricted to a tight cluster of infections along a single street [37]. These studies support an unrecognized and persistent burden of CHIKV transmission that currently escapes most arbovirus surveillance systems, and more research is urgently indicated to better understand patterns and mechanisms of CHIKV persistence.

The age-specific frequencies of CHIKV detection among febrile patients were not consistent with those of a simple endemic disease with equal rates of exposure and disability (in which case, we would have expected the incidence to be highest in children and decrease over time). The highest burden of CHIKV infection was observed in adults, which may reflect age-related differences in exposure patterns and / or differences in the frequencies and manifestations of CHIKV-related disease. Both of these factors may underlie the observed demographics of CHIKV-infected patients in this study. Notably, the incidence of arthralgia among febrile individuals with confirmed CHIKV infection (28.1%) was much lower than expected in this study, challenging the current dogma that joint pain is hallmark symptom of acute CHIKV [38, 39]. The possibility that a mis-alignment of the true signs and symptoms of acute CHIKV infection with the expected clinical picture may explain low detection rates of CHIKV infection through passive surveillance bears further investigation.

Prior to the most recent CHIKV epidemic in 2018–2019, the majority of transmission in southern Thailand was attributed to *Aedes albopictus*, as supported by specific mutations in E1 (A226V) and the apparent spatial restriction of CHIKV cases in Thailand to areas with higher densities of *Ae. albopictus* (i.e., southern Thailand). In contrast to *Aedes aegypti*, which tends to bite within and around the home, exposure to *Aedes albopictus* may have been more common for adult agricultural workers, for example those working in rubber plantations [40]. In 2018–2019, CHIKV isolates from across Thailand demonstrated mutations that are more suitable to transmission by *Aedes aegypti* (E1-A226, E1-K211E, and E2-V264A), which may be associated with significant shifts in the patterns of transmission and risk in the study area [8]; phylogenetic analyses indicate that this strain was similar to a strain from Bangladesh in 2017 [40].

The study is subject to limitations. Participants were derived from a passive surveillance program, where there may have been an unintentional bias toward enrolling individuals with presumed arboviral syndromes, or individuals from certain demographic groups or regions. Prospective cohort studies are needed to better understand the true burden of CHIKV infection and illness in the study area, and to capture the full clinical spectrum of disease. The rate of capture of RT-PCR-confirmed infections, relative to serological confirmation, was relatively low, precluding extensive phylogenetic analyses of viral persistence and likely reflecting delayed capture of cases. Supporting this explanation, individuals with RT-PCR- confirmed infections were significantly more likely to have presented earlier in the course of their illness than serologically-confirmed cases. Additionally, despite best efforts during sample transport and storage, hemolysis was noted in some samples which could have negatively impacted molecular detection of samples. An additional limitation is that a majority (308, 21%) of those identified as CHIKV positive were identified by HAI seroconversion with a low rate of RT-PCR positive individuals. Although RT-PCR negative, these seroconversions likely represent true cases as supported by PRNT on a random subset of 91 paired HAI seropositive and seronegative completed based on availability of paired samples. Among HAI seropositive individuals (n = 45), PRNT seroconversion was detected in 41 individuals and among HAI seronegative individuals (n = 46), negative PRNT was observed in 41 individuals, yielding a 9.89% disagreement rate. Intensified surveillance and heightened community and provider awareness regarding local persistence of CHIKV may improve viral capture in future studies.

This observational study of CHIKV in southern Thailand provides evidence of potential persistent circulation of CHIKV and highlights multiple critical areas for future research. It is likely that CHIKV continues to circulate at low levels in many regions of the world, escaping detection by current surveillance programs. An improved understanding of the ecology and epidemiology of the virus in various transmission settings is urgently needed, to include identification of suitable vectors, animal hosts, and human environments and behaviors most associated with risk. The intensified study of patterns and features of CHIKV persistence in locations suggested to be experiencing long-term transmission such as Thailand, India, and East Africa, for example, may provide unique opportunities to advance counter-measure development. For episodic pathogens such as CHIKV, intensified detection and study of persistent hot spots may hold the key to proactive preparation, rather than reactive attempts at mitigation, for future epidemics.

## Disclaimer

Material has been reviewed by the Walter Reed Army Institute of Research. There is no objection to its presentation and/or publication. The opinions or assertions contained herein are the private views of the author, and are not to be construed as official, or as reflecting true

views of the Department of the Army or the Department of Defense. The investigators have adhered to the policies for protection of human subjects as prescribed in AR 70–25.

## Supporting information

**S1 Fig. Chikungunya reported to the Ministry of Health in southern Thailand by province and year, 2008–2019.**
(TIFF)

**S2 Fig. Acute CHIKV infection and HAI titers.**
(TIFF)

**S3 Fig. HAI and PRNT Correlation.**
(TIFF)

**S4 Fig. Assay information.**
(TIFF)

## Author Contributions

**Conceptualization:** Darunee Buddhari.

**Data curation:** Aaron Farmer, Darunee Buddhari, Boonsri Charernmak, Butsaya Thaisomboonsuk.

**Formal analysis:** Kathryn B. Anderson, Butsaya Thaisomboonsuk, Tippa Wongstitwilairoong, Taweewun Hunsawong, Chonticha Klungthong, Piyawan Chinnawirotpisan.

**Funding acquisition:** Chonticha Klungthong, Stefan Fernandez.

**Investigation:** Darunee Buddhari, Thanaporn Hortiwakul, Boonsri Charernmak, Sarunyou Chusri.

**Methodology:** Kathryn B. Anderson, Darunee Buddhari, Thanaporn Hortiwakul, Boonsri Charernmak, Chonticha Klungthong, Piyawan Chinnawirotpisan.

**Project administration:** Aaron Farmer, Darunee Buddhari, Thanaporn Hortiwakul, Boonsri Charernmak, Sarunyou Chusri, Stefan Fernandez.

**Resources:** Taweewun Hunsawong, Sarunyou Chusri.

**Software:** Butsaya Thaisomboonsuk.

**Supervision:** Sarunyou Chusri.

**Validation:** Butsaya Thaisomboonsuk, Tippa Wongstitwilairoong, Taweewun Hunsawong.

**Visualization:** Butsaya Thaisomboonsuk, Tippa Wongstitwilairoong, Taweewun Hunsawong.

**Writing – original draft:** Aaron Farmer, Kathryn B. Anderson, Sarunyou Chusri, Stefan Fernandez.

**Writing – review & editing:** Aaron Farmer, Kathryn B. Anderson, Darunee Buddhari, Thanaporn Hortiwakul, Boonsri Charernmak, Tippa Wongstitwilairoong, Taweewun Hunsawong, Chonticha Klungthong, Piyawan Chinnawirotpisan, Sarunyou Chusri, Stefan Fernandez.

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
