## [Decision Letter · Decision Letter 0]

9 Apr 2024

Dear Dr. Farmer,

Thank you very much for submitting your manuscript "Continuous detection of Chikungunya Virus in a passive surveillance system in southern Thailand, 2012-2019" for consideration at PLOS Neglected Tropical Diseases. As with all papers reviewed by the journal, your manuscript was reviewed by members of the editorial board and by several independent reviewers. In light of the reviews (below this email), we would like to invite the resubmission of a significantly-revised version that takes into account the reviewers' comments. 

Please address all the comments by the 2 reviewers. In addition, the following comments should also be addressed:

1. HAI picked up more positive CHIKV compared to PCR. Is this normal? 

2. Have authors look at CHIKV outbreaks in the SE Asia region? This will influence the continuous transmission. 

3. This paper should also have data for 2008-2009 CHIKV outbreak. Otherwise, the 'inter-epidemic' is actually before the 2018-2019 epidemic. 

4.Ross and vaccine strain for viral assays. Are there any discrepancy compared to the current circulating strains? 

5. Fig S1, y axis is missing.

We cannot make any decision about publication until we have seen the revised manuscript and your response to the reviewers' comments. Your revised manuscript is also likely to be sent to reviewers for further evaluation.

Sincerely,

Yoke Fun Chan, PhD

Academic Editor

Mabel Carabali

Section Editor

Please address all the comments by the 2 reviewers. In addition, the following comments should also be addressed:

1. HAI picked up more positive CHIKV compared to PCR. Is this normal? 

2. Have authors look at CHIKV outbreaks in the SE Asia region? This will influence the continuous transmission. 

3. This paper should also have data for 2008-2009 CHIKV outbreak. Otherwise, the 'inter-epidemic' is actually before the 2018-2019 epidemic. 

4.Ross and vaccine strain for viral assays. Are there any discrepancy compared to the current circulating strains? 

5. Fig S1, y axis is missing.

Reviewer's Responses to Questions

**Key Review Criteria Required for Acceptance?**

**Methods**

-Are the objectives of the study clearly articulated with a clear testable hypothesis stated?

-Is the study design appropriate to address the stated objectives?

-Is the population clearly described and appropriate for the hypothesis being tested?

-Is the sample size sufficient to ensure adequate power to address the hypothesis being tested?

-Were correct statistical analysis used to support conclusions?

-Are there concerns about ethical or regulatory requirements being met?

Reviewer #1: Are the objectives of the study clearly articulated with a clear testable hypothesis stated? yes

-Is the study design appropriate to address the stated objectives? yes, except the use of PRNT and HAI. Authors describe to evaluate the correlation which is not stated in the objective. The method that take precedence when discrepancies occur should be mentioned. Authors did not describe whether if only acute specimens are available, these will also be tested.

line 140: was applied to quantify CHIKV neutralizing antibody titers in acute and140

convalescent serum specimens [30). Should be HAI titers?

line 146: Neutralizing antibody titers to CHIKV were measured in a randomly-selected subset of146acute specimens from study participants. Not acute and convalescent specimens?

-Is the population clearly described and appropriate for the hypothesis being tested? yes

-Is the sample size sufficient to ensure adequate power to address the hypothesis being tested? yes

-Were correct statistical analysis used to support conclusions? yes

-Are there concerns about ethical or regulatory requirements being met? no

Reviewer #2: (No Response)

**Results**

-Does the analysis presented match the analysis plan?

-Are the results clearly and completely presented?

-Are the figures (Tables, Images) of sufficient quality for clarity?

Reviewer #1: Does the analysis presented match the analysis plan? yes

-Are the results clearly and completely presented? yes. However, I would suggest authors to describe clearly, why when only acute specimens are available, they are not included in the study. how many % of subjects are positive for both PCR and serology (HAI or PRNT?), how many by PCR only and how many by serology only. Are there discrepancies between HAI and PRNT results? Are there any differences between clinical manifestation during epidemic and interepidemic? For non CHIKV group, would be better if the pathogens are described

-Are the figures (Tables, Images) of sufficient quality for clarity? yes

Reviewer #2: (No Response)

**Conclusions**

-Are the conclusions supported by the data presented?

-Are the limitations of analysis clearly described?

-Do the authors discuss how these data can be helpful to advance our understanding of the topic under study?

-Is public health relevance addressed?

Reviewer #1: Are the conclusions supported by the data presented? yes

-Are the limitations of analysis clearly described? yes

-Do the authors discuss how these data can be helpful to advance our understanding of the topic under study? yes

-Is public health relevance addressed? yes

Authors do not describe whether it is common that CHIKV infections do not result in arthralgia (only 15%). Is it the typical finding in interepidemic cases/ A separately analysis between the interepidemic and epidemic would be great.

Authors found all isolates that are sequenced are ECSA genotype. Is it common the the whole Thailand and SEA? A discussion regarding genotyping in the region and how it affect the clinical manifestations (usually more severe? will add the quality of the paper.

Reviewer #2: (No Response)

**Editorial and Data Presentation Modifications?**

Reviewer #1: Thank you for the opportunity to review this paper. This CHIKV surveillance results from South of Thailand is interesting for the community and well-written. I have several suggestions to improve the quality of the paper and provide more information for clinicians and researchers in SEA region.

Reviewer #2: (No Response)

**Summary and General Comments**

Reviewer #1: -

Reviewer #2: General comment

The transmission of CHIKV is unpredictable which classically manifest as explosive epidemics separated by variable interepidemic period. It remains unclear where CHIKV retreat during these inter-epidemic periods. By passive surveillance it was found that there is persistent circulation of CHIKV in the study area in patients with fever. 

Specific comments

1. First several lines of the Abstract and Author Summary is same. It should be revised.

2. Page 5, line 74: The US Food and Drug Administration has approved Ixchiq (VLA1553), the first vaccine for chikungunya [Chikungunya vaccine approved Nature Biotechnology volume 41, page1667 (2023)].

3. The investigation was done in patients presented with fever from 2012 to 2019. However, in 2017 no CHIKV was detected in patients. No attempt was taken to find during this period where the virus was in asymptomatic carriers or non-human primates. The objective of the study remains unsolved. 

4. Page 11, lines 201-202: The meaning is not clear, please rewrite.

5. Page 12, line 209: Is it incidence or prevalence?

6. Throughout the manuscript CHIKV epidemic was mentioned in several occasion. Please define CHIKV epidemic.

7. Page 14, lines 247-255: It is not clear why annual patterns of CHIKV and DENV was analysed.

8. Page 16, line 280: If there is no new findings in this study then what is the importance or impact this study carries?

9. Page 16, lines 295-296: How it escapes surveillance?

10. Page 17, line 319: Phylogenetic analysis only indicates that whether the strains are related or not. How the authors conclude that the strain was imported from Bangladesh?

PLOS authors have the option to publish the peer review history of their article (what does this mean?). If published, this will include your full peer review and any attached files.

Reviewer #1: No

Reviewer #2: No
---

## [Decision Letter · Decision Letter 1]

2 Aug 2024

Dear Dr. Farmer,

Thank you very much for submitting your manuscript "Continuous detection of Chikungunya Virus in a passive surveillance system in southern Thailand, 2012-2019" for consideration at PLOS Neglected Tropical Diseases. As with all papers reviewed by the journal, your manuscript was reviewed by members of the editorial board and by several independent reviewers. The reviewers appreciated the attention to an important topic. Based on the reviews, we are likely to accept this manuscript for publication, providing that you modify the manuscript according to the review recommendations. 

Please prepare and submit your revised manuscript within 15 days. If you anticipate any delay, please let us know the expected resubmission date by replying to this email. 

Sincerely,

Yoke Fun Chan, PhD

Academic Editor

Mabel Carabali

Section Editor

Reviewer's Responses to Questions

**Key Review Criteria Required for Acceptance?**

**Methods**

-Are the objectives of the study clearly articulated with a clear testable hypothesis stated?

-Is the study design appropriate to address the stated objectives?

-Is the population clearly described and appropriate for the hypothesis being tested?

-Is the sample size sufficient to ensure adequate power to address the hypothesis being tested?

-Were correct statistical analysis used to support conclusions?

-Are there concerns about ethical or regulatory requirements being met?

Reviewer #1: It would be helpful if authors prepare a diagram to show the flow of diagnostic assay to make it easier for readers to understand the methods of this surveillance study

Reviewer #3: (No Response)

**Results**

-Does the analysis presented match the analysis plan?

-Are the results clearly and completely presented?

-Are the figures (Tables, Images) of sufficient quality for clarity?

Reviewer #1: 1. It is quite surprising that there is no difference in the frequency of myalgia and arthralgia in CHIK and non-CHIK groups. Also, it makes sense that the frequency during epidemic seasons were higher than interepidemic (although only for myalgia, as the p value in arthralgia is 0.045). I think you should mention the tendency of arthralgia to be more frequent in the epidemic group, although not significant. Readers do not have to go to the supplement to find out the p value. In my opinion it is important to discuss these findings (myalgia/arthralgia in CHIK/non CHIK and in epidemic/inter epidemic, as these clinical findings are known to be the prominent symptoms of CHIKV infections.

2. Authors have discussed why you chose HI instead of PRNT for diagnosing (more practical). It will be informative for readers if you also add why you did not choose ELISA method, which is even more practical than HI, and has been widely used?

3. It is important for authors to discuss why there were many specimens during acute illness that were missed by RT-PCR (average negative by RT-PCR was 2.6 days after onset of illness), suggesting that there were quite many specimens obtained before 2.6 days negative by RT-PCR. 

4. Several inconsistencies between results and method: e.g. a) in the method section, the max day of convalescent specimens is 21 days apart, but in the result section is 35 days (line 204). b) in the method, PRNT was conducted in acute specimens randomly chosen, but you compared the results of HI and PRNT of the convalescent specimens. c) line 141-142: HI was performed when acute and convalescent specimens are available, but the previous sentence stated when convalescent specimens are not available, they are not included in the study. d) line 148-150, and line 176-177 PCR was conducted in acute and convalescent specimens?

5. In the response to the reviewers, authors explain that cases confirmed by RT-PCR only was 90 (6.1%). Does it mean, HI results were not consistent with RT-PCR results, or they were not tested for HI?. Authors also did not response to the question from reviewers regarding the discrepancies of PRNT and HI. Instead, authors explained that the correlation was 0.817. Does it suggest that they are some discrepancies? When it occurred (e.g. HI positive, but PRNT negative), what the diagnostic decision was?

6. Regarding the proportion of dengue cases (17.5%), it is not clear the denominator that is used. Authors say from all cases (>2127 or 1473 subjects). Without the denominator, it is hard to understand this sentence:" Acute DENV infection was confirmed by RT-PCR in 258 (17.5%) of all participants presenting with acute febrile illness. Among these, 3 were also RT-PCR-positive for CHIKV, for a PCR-confirmed coinfection rate of 1.2%"

Was dengue confirmed by RT-PCR only, or also serological assay?. Can we compare the results from dengue and chikungunya cases? 

7. It is hard to understand this sentence: 

In the text: Among the 1473 individuals that presented with febrile illness from 2012-2019 and were eligible for inclusion in this analysis, 401 (27.3%) had confirmed acute CHIKV infection by 202 molecular and/or serological methods (PCR and/or acute HAI sample positive); of these 90 of 1473 (6.1%) of individuals had detectable CHIKV RNA in acute specimens by RT-PCR. 

In the response to reviewers: 

a) PCR AND Serology- 45 (3.1%)

b) PCR only- 90 (6.1%)

c) Serology only- 356 (24.2%)

8. As I already suggest in the method section, it will be easier if authors provide a figure that explains the flow of the tests and the results as well.

Reviewer #3: (No Response)

**Conclusions**

-Are the conclusions supported by the data presented?

-Are the limitations of analysis clearly described?

-Do the authors discuss how these data can be helpful to advance our understanding of the topic under study?

-Is public health relevance addressed?

Reviewer #1: Conclusions are supported by data

Limitations are described

The contributions to public health are also discussed

Reviewer #3: (No Response)

**Editorial and Data Presentation Modifications?**

Reviewer #1: I recommend 'Minor Revision' since there are several sections should be modified to enhance clarity.

Reviewer #3: (No Response)

**Summary and General Comments**

Reviewer #1: Thank you for the opportunity to review this important study. I have a few suggestions for improvement and clarification.

Reviewer #3: This manuscript by Farmer and colleagues presents eight years of longitudinal passive surveillance for chikungunya and dengue in southern Thailand. The authors demonstrate evidence for near-continual circulation of CHIKV, with expected increases and decreases in percent positive among enrolled cases by year. Perhaps the most striking finding is the variability in proportion of PCR-positive versus serology-positive chikungunya cases, in particular between epidemic and inter-epidemic periods. This observation is surprising; however, the most important conclusions of the manuscript are mirrored whether or not the serology-positives cases are included. In addition, the authors present sufficient evaluation of their HAI method to render the findings convincing. Overall, this is an important manuscript that presents findings that are quite relevant to the field.

My only minor suggestion for revision is that "chikungunya" should never be capitalized (unless it begins a sentence), as the word originates not from a specific location but rather from an indigenous word referring to the illness.

PLOS authors have the option to publish the peer review history of their article (what does this mean?). If published, this will include your full peer review and any attached files.

Reviewer #1: Yes: Herman Kosasih

Reviewer #3: Yes: Tyler M. Sharp

Figure Files:

Data Requirements:

Reproducibility:

References

---

## [Editor Report · Decision Letter 2]

8 Nov 2024

Dear Dr. Farmer,

Thank you very much for submitting your manuscript "Continuous detection of Chikungunya Virus in a passive surveillance system in southern Thailand, 2012-2019" for consideration at PLOS Neglected Tropical Diseases. As with all papers reviewed by the journal, your manuscript was reviewed by members of the editorial board and by several independent reviewers. The reviewers appreciated the attention to an important topic. Based on the reviews, we are likely to accept this manuscript for publication, providing that you modify the manuscript according to the review recommendations. 

The manuscript is now close to acceptance but requires a few minor edits. 

1, Please address the the Q5 by reviewer in the manuscript, show the numbers, and address the discrepancy between HAI and PRNT in limitation. 

1. Please check all the numbers are using the denominator 1466 instead of 1473. The reply to reviewer Q6 still use 1473.

Sincerely,

Yoke Fun Chan, PhD

Academic Editor

Mabel Carabali

Section Editor

Figure Files:

Data Requirements:

Reproducibility:

References

---

## [Editor Report · Decision Letter 3]

10 Dec 2024

Dear Dr. Farmer,

We are pleased to inform you that your manuscript 'Continuous detection of Chikungunya Virus in a passive surveillance system in southern Thailand, 2012-2019' has been provisionally accepted for publication in PLOS Neglected Tropical Diseases.

Best regards,

Yoke Fun Chan, PhD

Academic Editor

Mabel Carabali

Section Editor

Shaden Kamhawi

co-Editor-in-Chief

Paul Brindley

co-Editor-in-Chief

---

## [Editor Report · Acceptance letter]

20 Dec 2024

Dear Dr. Farmer,

We are delighted to inform you that your manuscript, "Continuous detection of Chikungunya Virus in a passive surveillance system in southern Thailand, 2012-2019," has been formally accepted for publication in PLOS Neglected Tropical Diseases.

Best regards,

Shaden Kamhawi

co-Editor-in-Chief

Paul Brindley

co-Editor-in-Chief
